# SIRT2 inhibition protects against cardiac hypertrophy and ischemic injury

Xiaoyan Yang[†], Hsiang-Chun Chang[†], Yuki Tatekoshi[†], Amir Mahmoodzadeh, Maryam Balibegloo, Zeinab Najafi, Rongxue Wu, Chunlei Chen, Tatsuya Sato, Jason Shapiro, Hossein Ardehali*

Feinberg Cardiovascular and Renal Research Institute, Northwestern University School of Medicine, Chicago, United States

**Abstract** Sirtuins (SIRT) exhibit deacetylation or ADP-ribosyltransferase activity and regulate a wide range of cellular processes in the nucleus, mitochondria, and cytoplasm. The role of the only sirtuin that resides in the cytoplasm, SIRT2, in the development of ischemic injury and cardiac hypertrophy is not known. In this paper, we show that the hearts of mice with deletion of *Sirt2* (*Sirt2⁻/⁻*) display improved cardiac function after ischemia-reperfusion (I/R) and pressure overload (PO), suggesting that SIRT2 exerts maladaptive effects in the heart in response to stress. Similar results were obtained in mice with cardiomyocyte-specific *Sirt2* deletion. Mechanistic studies suggest that SIRT2 modulates cellular levels and activity of nuclear factor (erythroid-derived 2)-like 2 (NRF2), which results in reduced expression of antioxidant proteins. Deletion of *Nrf2* in the hearts of *Sirt2⁻/⁻* mice reversed protection after PO. Finally, treatment of mouse hearts with a specific SIRT2 inhibitor reduced cardiac size and attenuates cardiac hypertrophy in response to PO. These data indicate that SIRT2 has detrimental effects in the heart and plays a role in cardiac response to injury and the progression of cardiac hypertrophy, which makes this protein a unique member of the SIRT family. Additionally, our studies provide a novel approach for treatment of cardiac hypertrophy and injury by targeting SIRT2 pharmacologically, providing a novel avenue for the treatment of these disorders.

*For correspondence:
h-ardehali@northwestern.edu

†These authors contributed equally to this work

Competing interest: The authors declare that no competing interests exist.

## Editor's evaluation

In this manuscript, the authors examine the role of Sirt2 on cardiac hypertrophy by using 2 in-vivo models- systemic KO of Sirt2 and cardiac specific KO of Sirt 2. They have shown that Sirt2 is important for development of heart failure and cardiac hypertrophy. Mechanistically, the authors show that Sirt2 regulates NRF2 and that deletion of Sirt2 is protective through stabilization and increased nuclear translocation of NRF2.

## Introduction

Sirtuin (SIRT) family of proteins comprise class III of histone deacetylases. SIRTs require NAD⁺ to carry out their enzymatic reaction, and have been implicated in a wide range of cellular processes including aging, apoptosis, response to stress and inflammation, control of energy efficiency, circadian clocks, and mitochondrial biogenesis (*Baur et al., 2012*; *Preyat and Leo, 2013*). In mammals, seven sirtuins (SIRT1-7) have been identified, which are categorized according to their subcellular localization to the nucleus (SIRT1, -6, and -7), mitochondria (SIRT3, -4, and -5), and cytoplasm (SIRT2). SIRT1–3 have a robust deacetylation activity, while SIRT4 is reported to display ADP-ribosyltransferase activity. SIRT5 may function as a protein desuccinylase and demalonylase, and SIRT6 and SIRT7 display weak deacetylase activity (*Du et al., 2011*; *Haigis et al., 2006*; *Michishita et al., 2005*; *Pan et al., 2011*).

A number of SIRTs have been studied in the heart (for review, please see *Matsushima and Sadoshima, 2015*). The effects of SIRT1 in the heart are complex. *Sirt1* deletion protects against pressure overload (PO)-induced cardiac hypertrophy (*Oka et al., 2011*; *Sundaresan et al., 2011*); however, low-level overexpression of SIRT1 in the heart attenuates age-associated cardiac hypertrophy, fibrosis, and cardiac dysfunction, while high-level overexpression of SIRT1 increases these pathological effects (*Alcendor et al., 2007*). In the setting of ischemia-reperfusion (I/R), SIRT1 exerts protective effects: *Sirt1* knockout (KO) in the heart increases I/R-induced injury, while its overexpression protects against I/R-induced injury (*Hsu et al., 2010*). Thus, it appears that the effects of SIRT1 on cardiac response to stress are dependent on its expression levels as well as the context of injury. SIRT3 has also been studied in the heart and shown to protect against both cardiac hypertrophy and I/R injury (*Porter et al., 2014*; *Sundaresan et al., 2009*), while SIRT6 KO mice exhibit cardiac hypertrophy (*Sundaresan et al., 2012*). Recent studies have assessed the role of SIRT2 in the heart. One study showed that deletion of *Sirt2* reduces AMPK activation and increases age-related and angiotensin II-mediated cardiac hypertrophy (*Tang et al., 2017*), while another showed that advanced glycation end products and its receptor promote diabetic cardiomyopathy through suppression of SIRT2, however, KO mice were not used for these studies (*Yuan et al., 2015*). Another study showed that *Sirt2* deficiency increases nuclear localization of NFATc2 and its transcription activity, and that NFAT inhibition rescues the cardiac dysfunction in mice with *Sirt2* deletion (*Sarikhani et al., 2018*).

Nuclear factor (erythroid-derived 2)-like 2 (NRF2) is a transcription factor that activates a number of cytoprotective genes, including antioxidative enzymes (*Tufekci et al., 2011*). Under normal conditions, NRF2 resides in the cytoplasm, and is degraded primarily through its interaction with Keap1 (kelch-like ECH-associated protein 1), which also serves as a bridge between NRF2 and cullin 3-ubiquitination complex (*Tufekci et al., 2011*). Under oxidative stress, NRF2 escapes degradation, translocates into the nucleus, and binds to antioxidant response elements in the promoter of a number of genes (*Kaspar et al., 2009*). NRF2 acetylation is decreased with SIRT1 overexpression (*Kawai et al., 2011*); however, an association between SIRT1 and NRF2 has not been demonstrated, and the functional consequences of NRF2 deacetylation have not been studied. NRF2 KO mice developed cardiac hypertrophy and heart failure (HF) after trans-aortic constriction (TAC) (*Li et al., 2009*), indicating the NRF2 is protective against cardiac stress. We recently showed that SIRT2 mediates NRF2 deacetylation in the liver cells and its translocation in the nucleus to regulate antioxidant genes (*Yang et al., 2017*).

In this paper, we show that SIRT2 plays a detrimental role in the heart in response to injury, in contrast to a previously published report (*Tang et al., 2017*). Mechanistically, deletion of *Sirt2* is protective through stabilization and increased nuclear translocation of NRF2, leading to increased expression of antioxidant genes. Finally and most importantly, we show that pharmacological inhibition of SIRT2 protects the heart against the development of cardiac hypertrophy, opening potential treatment for this disorder.

## Results
### SIRT2 is expressed in the heart and its levels are elevated in HF
We first showed that SIRT2 is expressed in the heart (*Figure 1—figure supplement 1A*) and in H9c2 cardiomyoblasts (*Figure 1—figure supplement 1B*) at relatively high levels. We also found that SIRT2 expression was higher in the hearts of mice 4 weeks after TAC compared to sham (*Figure 1A*), while the levels of other sirtuin family members with major deacetylation activity that have been studied in the heart (i.e., SIRT1, SIRT3, and SIRT6) were not different. Additionally, we noted a significant increase in the levels of SIRT2 in the explanted hearts from end-stage HF patients with dilated cardiomyopathy (*Figure 1B*). We also assessed the levels of SIRT2 in explanted hearts from patients with ischemic cardiomyopathy and showed that SIRT2 is increased in these hearts (*Figure 1C*). These results indicate that SIRT2 levels are increased in HF and ischemic injury.

### *Sirt2* deficiency preserves cardiac function in response to PO and I/R injury
We first used mice with global deletion of *Sirt2* KO (*Sirt2*-/-) for our studies. We assessed whether *Sirt2* deletion affects the levels of other sirtuin family members in the heart. *Sirt2*-/- hearts displayed no change in other sirtuin family members at the mRNA level, and no change in protein levels of SIRT1,

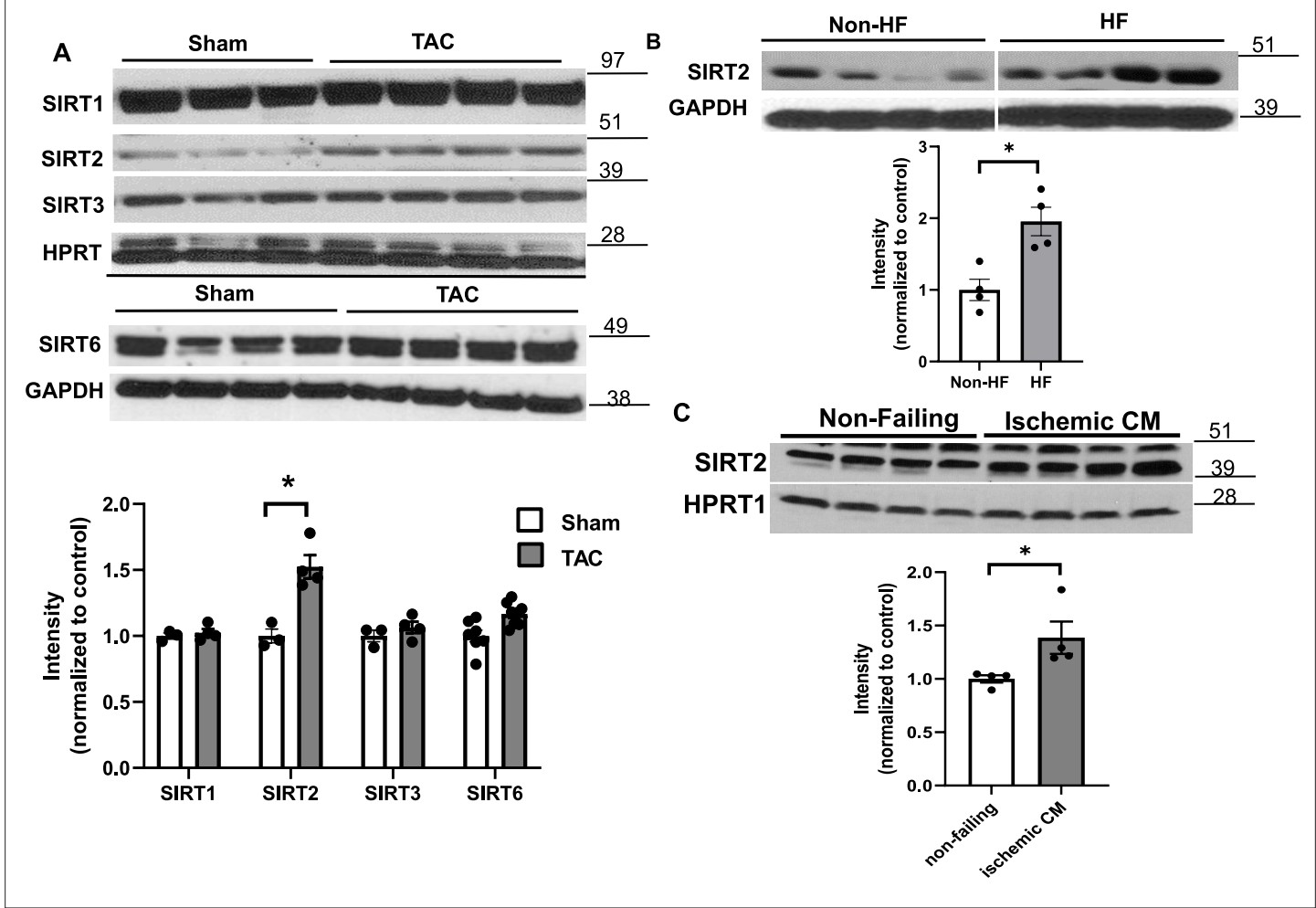

**Figure 1.** SIRT2 is upregulated in heart failure (HF). (**A**) SIRT1, SIRT2, SIRT3, and SIRT6 in mouse hearts after trans-aortic constriction (TAC). (**B**) SIRT2 in human hearts from healthy patients and patients with dilated cardiomyopathy. (**C**) SIRT2 protein levels in the hearts of control individual and patients with ischemic heart failure. *p<0.05 by Student's t-test. Data presented as mean ± SEM.

The online version of this article includes the following source data and figure supplement(s) for figure 1:

**Source data 1.** SIRT1, -2, -3, and -6 after sham and trans-aortic constriction (TAC) surgery as shown in *Figure 1A*.

**Source data 2.** SIRT2 in non-failing and failing human hearts as shown in *Figure 1B*.

**Source data 3.** SIRT2 in non-failing and ischemic human hearts as shown in *Figure 1C*.

**Source data 4.** Full gels for *Figure 1A–C*.

**Source data 5.** Full gels for *Figure 1A–C* unedited.

**Source data 6.** Full gels for *Figure 1A–C* unedited.

**Figure supplement 1.** SIRT2 protein in different mouse tissues, including the heart (**A**), and in various cell lines, including H9c2 cells (**B**).

**Figure supplement 1—source data 1.** Full gels for *Figure 1—figure supplement 1*.

**Figure supplement 1—source data 2.** Full gels for *Figure 1—figure supplement 1* unedited.

SIRT3, or SIRT6 (sirtuins with major deacetylation activity) was detected (*Figure 2—figure supplement 1*). We then assessed whether *Sirt2* deletion protects against PO. *Sirt2$^{-/-}$* mice displayed normal cardiovascular parameters at baseline and no overt phenotype. However, in response to TAC, *Sirt2$^{-/-}$* mice displayed improved cardiac function than littermate controls, as assessed by fractional shortening (FS) and ejection fraction (EF) (*Figure 2A and B*). Additionally, *Sirt2$^{-/-}$* mice displayed evidence of less cardiac hypertrophy, as evidenced by lower interventricular septal (IVS) thickness on echocardiography (*Figure 2C*), and reduced cardiac size and heart weight to body weight ratio on gross examination (*Figure 2D and E*). Histological examination of the hearts also showed smaller cardiomyocytes

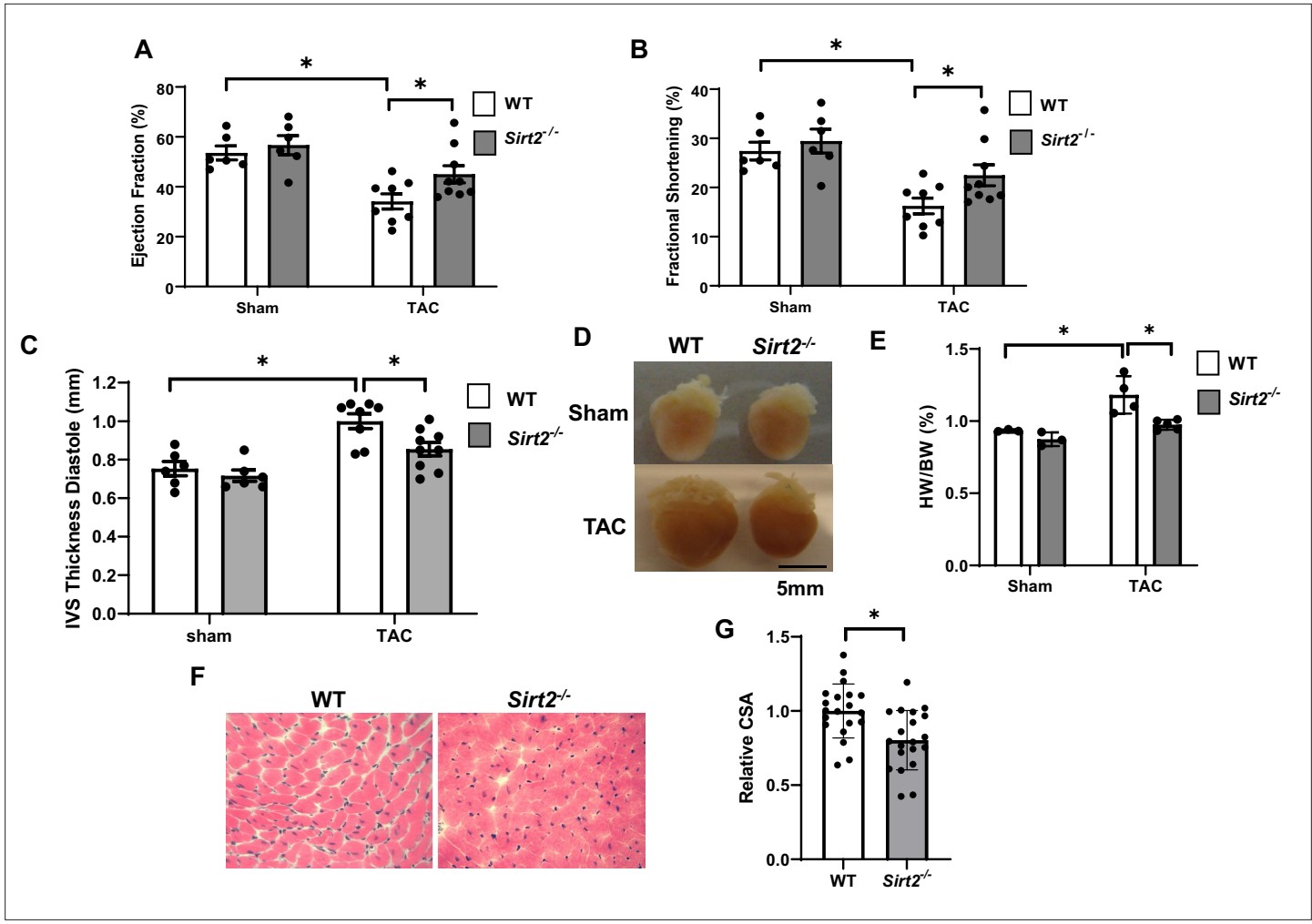

**Figure 2.** *Sirt2* deficiency protects the heart against cardiac dysfunction after trans-aortic constriction (TAC). *Sirt2⁻/⁻* and wild-type (WT) littermates were subjected to TAC and ejection fraction (EF) (**A**), fractional shortening (FS) (**B**), and interventricular septal thickness during diastole (**C**) were assessed 4 weeks later (N=6–9). (**D–F**) Representative hearts (**D**), HW/BW (**E**) (N=3–5), H&E staining, (**F**) and the summary of cross-sectional area of cardiomyocytes (**G**) in WT and *Sirt2⁻/⁻* hearts (N=20 cardiomyocytes), *p<0.05 by one-way ANOVA and post hoc Tukey analysis (**A, B, C, and E**) and unpaired Student's t-test (**G**). Bars represent group mean.

The online version of this article includes the following source data and figure supplement(s) for figure 2:

**Source data 1.** Ejection fraction (EF) in wild-type (WT) and *Sirt2⁻/⁻* mice after sham or trans-aortic constriction (TAC) as shown in ***Figure 2A***.

**Source data 2.** Fractional shortening (FS) in wild-type (WT) and *Sirt2⁻/⁻* mice after sham or trans-aortic constriction (TAC) as shown in ***Figure 2B***.

**Source data 3.** Interventricular septal (IVS) thickness diastole in wild-type (WT) and *Sirt2⁻/⁻* mice after sham or trans-aortic constriction (TAC) as shown in ***Figure 2C***.

**Source data 4.** HW/BW in wild-type (WT) and *Sirt2⁻/⁻* mice after sham or trans-aortic constriction (TAC) as shown in ***Figure 2E***.

**Source data 5.** CSA in wild-type (WT) and *Sirt2⁻/⁻* hearts as shown in ***Figure 2G***.

**Figure supplement 1.** Expression of protein (**A**) and mRNA (**B**) of other sirtuin family members in the hearts of *Sirt2⁻/⁻* mice.

**Figure supplement 1—source data 1.** mRNA levels of sirtuin proteins in wild-type (WT) and *Sirt2⁻/⁻* hearts as shown in ***Figure 2—figure supplement 1***.

**Figure supplement 1—source data 2.** Full gels for ***Figure 2—figure supplement 1***.

**Figure supplement 1—source data 3.** Full gels for ***Figure 2—figure supplement 1*** unedited.

in *Sirt2⁻/⁻* hearts after PO, as assessed by H&E staining (***Figure 2F and G***). These data indicate that deletion of *Sirt2* results in protection of the heart against PO with improved cardiac function and less cardiac hypertrophy.

To better assess the role of SIRT2 in the development of HF and ischemic damage, we then studied the effects of *Sirt2* deletion in the heart on the response to I/R. We subjected *Sirt2⁻/⁻* and their littermate

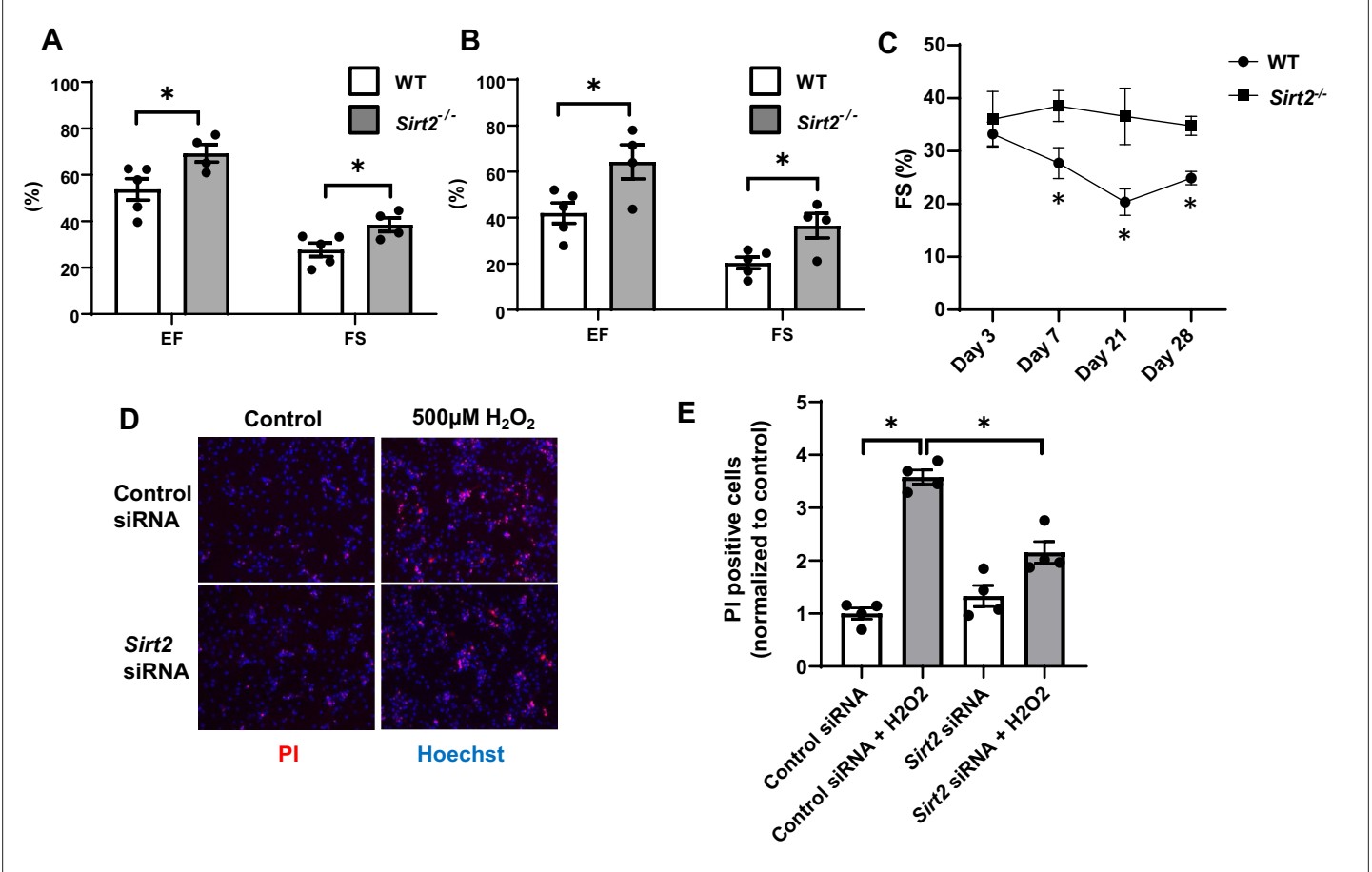

**Figure 3.** Hearts from *Sirt2*-/- mice are protected against ischemia-reperfusion (I/R) injury. Ejection fraction (EF) and fractional shortening (FS) in wild-type (WT) and *Sirt2*-/- mice 7 (**A**) and 21 days (**B**) after I/R (N=4–5). (**C**) Time course of FS in *Sirt2*-/- mice after I/R injury (N=4–5). (**D, E**) Cell death assessed by propidium iodide (PI), in neonatal rat cardiomyocyte (NRCM) treated with control or *Sirt2* siRNA and with 500 μM of H$_2$O$_2$. *p<0.05 by ANOVA for all panels expect for panel C, where Student's t-test was used for comparison between the two time points. Bars represent mean (**A, B**), and data presented as mean ± SEM.

The online version of this article includes the following source data for figure 3:

**Source data 1.** Ejection fraction (EF) and fractional shortening (FS) in wild-type (WT) and *Sirt2*-/- mice after ischemia-reperfusion (I/R) as shown in *Figure 3A*.

**Source data 2.** Ejection fraction (EF) and fractional shortening (FS) in wild-type (WT) and *Sirt2*-/- mice after ischemia-reperfusion (I/R) as shown in *Figure 3B*.

**Source data 3.** Time course of fractional shortening (FS) in wild-type (WT) and *Sirt2*-/- mice after ischemia-reperfusion (I/R) as shown in *Figure 3C*.

**Source data 4.** Propidium iodide (PI) positive cells as shown in *Figure 3E*.

wild-type (WT) controls to I/R and cardiac function was assessed after 7 and 21 days. At both time points, EF and FS were significantly higher in *Sirt2*-/- mice compared to controls (*Figure 3A and B*). Time course of cardiac assessment showed that while FS was comparable between WT and *Sirt2*-/- on day 3, it quickly deteriorated in WT mice, consistent with transition into HF, while *Sirt2*-/- mice maintained their cardiac function (*Figure 3C*). To further support these findings, we assessed the effects of *Sirt2* modulation on cell death in response to H$_2$O$_2$ in neonatal rat cardiomyocytes (NRCMs) treated with control or *Sirt2* siRNA by measuring propidium iodide (PI) positive cells. We found that cells with *Sirt2* knockdown (KD) displayed improved cell viability in response to H$_2$O$_2$ (*Figure 3D and E*). Overall, these results indicate that SIRT2 exerts detrimental effects in the heart in response to PO and I/R, and that its deletion leads to protective effects.

The experiments in *Figures 2 and 3* were conducted in mice with global deletion of *Sirt2*. To confirm a role for SIRT2 in cardiomyocyte response to injury, we then generated cardiac-specific

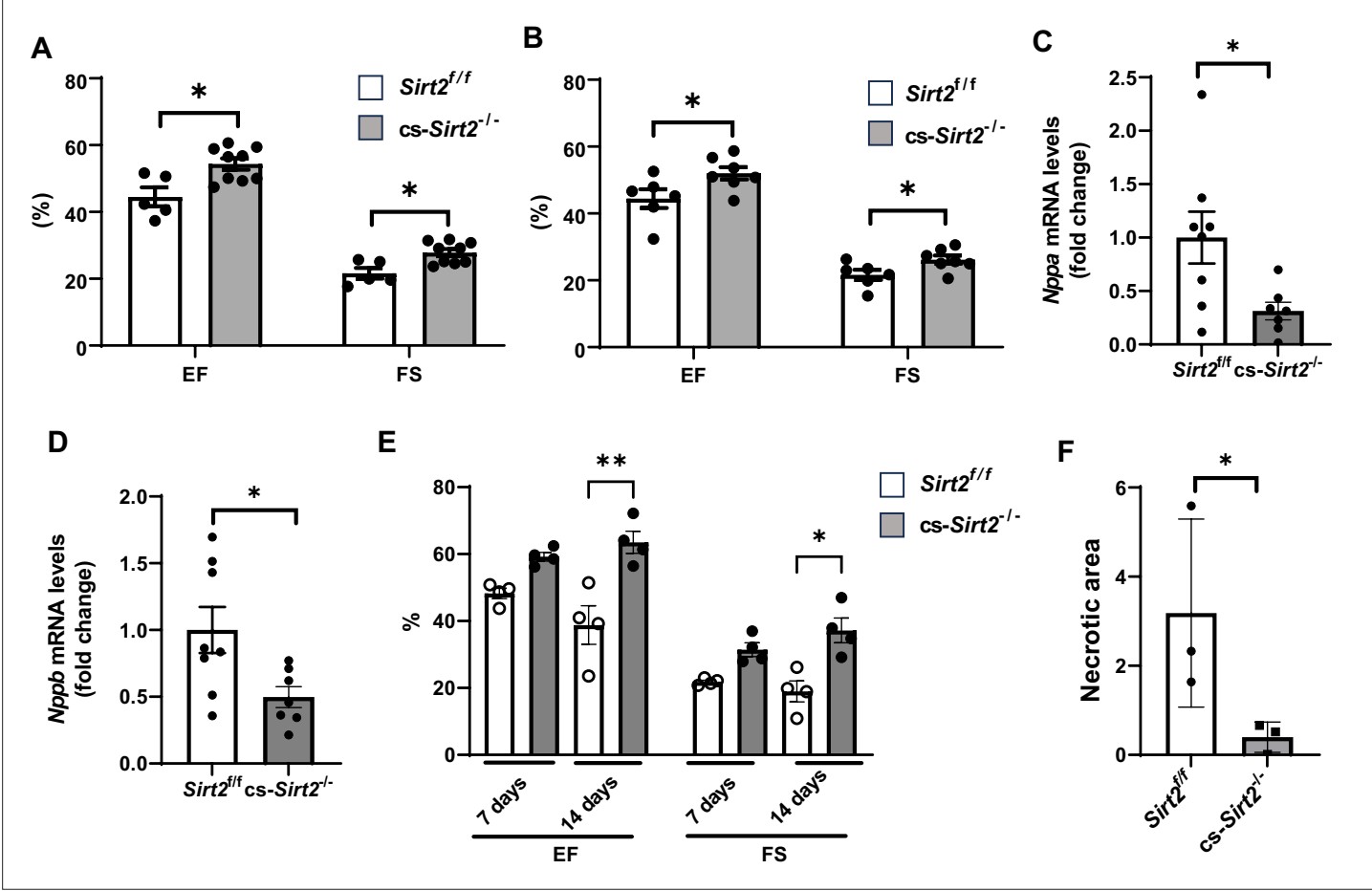

**Figure 4.** cs-*Sirt2*⁻/⁻ hearts are protected against trans-aortic constriction (TAC) and ischemia-reperfusion (I/R). Ejection fraction (EF) and fractional shortening (FS) in *Sirt2*^f/f and cs-*Sirt2*⁻/⁻ mice 7 (**A**) and 14 days (**B**) after TAC (N=5–9). (**C,D**) mRNA levels of *Anf* (**C**) and *Bnp* (**D**) in the hearts of *Sirt2*^f/f and cs-*Sirt2*⁻/⁻ mice 4 weeks after TAC (N=7–8). (**E**) EF and FS in *Sirt2*^f/f and cs-*Sirt2*⁻/⁻ mice 7 and 14 days after I/R (N=4). (**F**) Necrotic area (representing the degree of ischemic damage) in *Sirt2*^f/f and cs-*Sirt2*⁻/⁻ mice 14 days after MI. *$p<0.05$ by ANOVA for panels A and B, and Student's t-test was used for panels C and D. Data are presented as mean ± SEM.

The online version of this article includes the following source data and figure supplement(s) for figure 4:

**Source data 1.** Ejection fraction (EF) and fractional shortening (FS) in *Sirt2*^f/f and cs-*Sirt2*⁻/⁻ mice 7 days after ischemia-reperfusion (I/R) as shown in *Figure 4A*.

**Source data 2.** Ejection fraction (EF) and fractional shortening (FS) in *Sirt2*^f/f and cs-*Sirt2*⁻/⁻ mice 14 days after ischemia-reperfusion (I/R) as shown in *Figure 4B*.

**Source data 3.** *Nppa* mRNA in *Sirt2*^f/f and cs-*Sirt2*⁻/⁻ hearts as shown in *Figure 4C*.

**Source data 4.** *Nppb* mRNA in *Sirt2*^f/f and cs-*Sirt2*⁻/⁻ hearts as shown in *Figure 4D*.

**Source data 5.** Echo parameters in *Sirt2*^f/f and cs-*Sirt2*⁻/⁻ hearts as shown in *Figure 4E*.

**Figure supplement 1.** SIRT1, SIRT3, and SIRT2 protein in the hearts of cs-*Sirt2*⁻/⁻ mice.

**Figure supplement 1—source data 1.** Uncropped gels for *Figure 4—figure supplement 1*.

**Figure supplement 1—source data 2.** Uncropped gels for *Figure 4—figure supplement 1* unedited.

**Figure supplement 2.** cs-*Sirt2*⁻/⁻ hearts from female mice are protected against trans-aortic constriction (TAC).

**Figure supplement 2—source data 1.** Ejection fraction (EF) and fractional shortening (FS) in female wild-type (WT) and cs-Sirt2⁻/⁻ mice 7 and 14 days after TAC as shown in *Figure 4—figure supplement 2*.

Sirt2 KO mice (cs-*Sirt2*⁻/⁻) by crossing *Sirt2* floxed mice with αMHC-Cre mice. We confirmed lack of SIRT2 expression and no change in SIRT1 and SIRT3 in the hearts of cs-*Sirt2*⁻/⁻ mice (*Figure 4— figure supplement 1*). The cs-*Sirt2*⁻/⁻ mice were then subjected to TAC with littermate Cre negative mice as control, and cardiac functions (EF and FS) were assessed at 1 and 2 weeks after injury. At

both time points, cs-*Sirt2*[-/-] mice displayed improved cardiac function compared to WT controls (*Figure 4A and B*). Consistent with these data, HF markers in the heart, including *Nppa* and *Nppb*, were significantly lower in cs-*Sirt2*[-/-] mice after TAC (*Figure 4C and D*). To determine whether these effects are gender specific, we also preformed TAC in female *Sirt2*[f/f] and cs-*Sirt2*[-/-] mice and demonstrated that they are also protected against TAC at 7 and 14 days (*Figure 4—figure supplement 2*). Finally, the hearts of cs-*Sirt2*[-/-] mice are also protected against I/R, as their function was significantly higher (*Figure 4E*) and degree of ischemic damage was lower (*Figure 4F*) than *Sirt2*[f/]f mice 14 days after I/R.

## SIRT2 deacetylates NRF2 resulting in decreased transcriptional activity in the heart

We previously showed that SIRT2 deacetylates NRF2 protein in the liver and alters iron release from hepatocytes (*Yang et al., 2017*). Since deacetylation of NRF2 leads to protein destabilization and NRF2 regulates the expression of many antioxidant genes, we hypothesized that the mechanism for the protective effects of Sirt2 deletion in response to stress is through decreased NRF2 deacetylation and degradation, resulting in increased expression of antioxidant proteins. To test this hypothesis, we first assessed whether there is physical interaction between SIRT2 and NRF2 in the heart. Co-immunoprecipitation (IP) experiments showed that SIRT2 interacts with NRF2 in the heart of WT mice (*Figure 5A*).

We then measured acetylation levels of NRF2 in WT and *Sirt2*[-/-] hearts, and showed that NRF2 acetylation levels are increased within *Sirt2*[-/-] hearts (*Figure 5B*). We then assessed whether deacetylation of NRF2 alters its levels in the cardiac cells, as shown before in the liver (*Yang et al., 2017*). Treatment of NRCMs with *Sirt2* siRNA resulted in increased NRF2 protein levels compared with control siRNA, indicating that SIRT2 leads to a reduction in the levels of NRF2 protein (*Figure 5C*). Since NRF2 protein levels are higher in *Sirt2*[-/-] hearts, we next assessed whether SIRT2 alters the stability of NRF2 protein. NRF2 levels were significantly lower starting at 60 min after treatment with the protein synthesis inhibitor cycloheximide (CHX), leading to almost complete degradation at 120 min in cells treated with control siRNA. However, we noted no change in NRF2 protein levels in cells treated with *Sirt2* siRNA (*Figure 5D*). These data indicate that SIRT2 binds to NRF2, and its deacetylation leads to the instability and degradation of NRF2.

NRF2 is a transcription factor and upon activation, translocates into the nucleus to exert its transcriptional activity (*Hybertson et al., 2011*). Thus, we measured nuclear level of NRF2 and found it to be increased in NRCMs with *Sirt2* KD (*Figure 5E*). Since the increase in nuclear levels of NRF2 suggests possibly higher transcriptional activity of the protein, we next assessed the effects of *Sirt2* modulation on NRF2 transcriptional activity in H9c2 cells treated with lentivirus expressing either control or SIRT2 lentivirus. Consistent with its increased nuclear levels, SIRT2 overexpression in H9c2 cells resulted in lower levels of known NRF2 target genes (*Figure 5F–H*). However, the mRNA levels of non-NRF2 targeted antioxidant genes were not affected by SIRT2 overexpression (*Figure 5—figure supplement 1*). We confirmed the data in mouse HL-1 atrial cell line and showed an increase in NRF2 protein with *Sirt2* KD and a decrease in NRF2 target proteins with overexpression of SIRT2 (*Figure 5—figure supplement 2*).

Since our data suggest a role for SIRT2 in the regulation of NRF2-mediated expression of antioxidant genes, we next assessed whether SIRT2 has an effect on reactive oxygen species (ROS) production. NRCMs treated with *Sirt2* siRNA displayed less ROS levels after treatment with $H_2O_2$ (*Figure 5—figure supplement 3*), further supporting a role for SIRT2 in regulating oxidative state of cardiomyocytes.

## *Sirt2/Nrf2* double KO mice display more cardiac damage after I/R compared to *Sirt2*[-/-] mice

Our results thus far demonstrate that NRF2 is a target of SIRT2 and that SIRT2 regulates NRF2 acetylation and protein levels. To determine whether the protective effects of SIRT2 are mediated through NRF2, we generated *Sirt2/Nrf2* double KO mice and subjected the mice to I/R injury. The *Sirt2/Nrf2* double KO mice displayed reduced EF and FS compared to *Sirt2*[-/-] mice (*Figure 6A and B*), indicating that deletion of *Nrf2* reverses the protective effects of SIRT2.

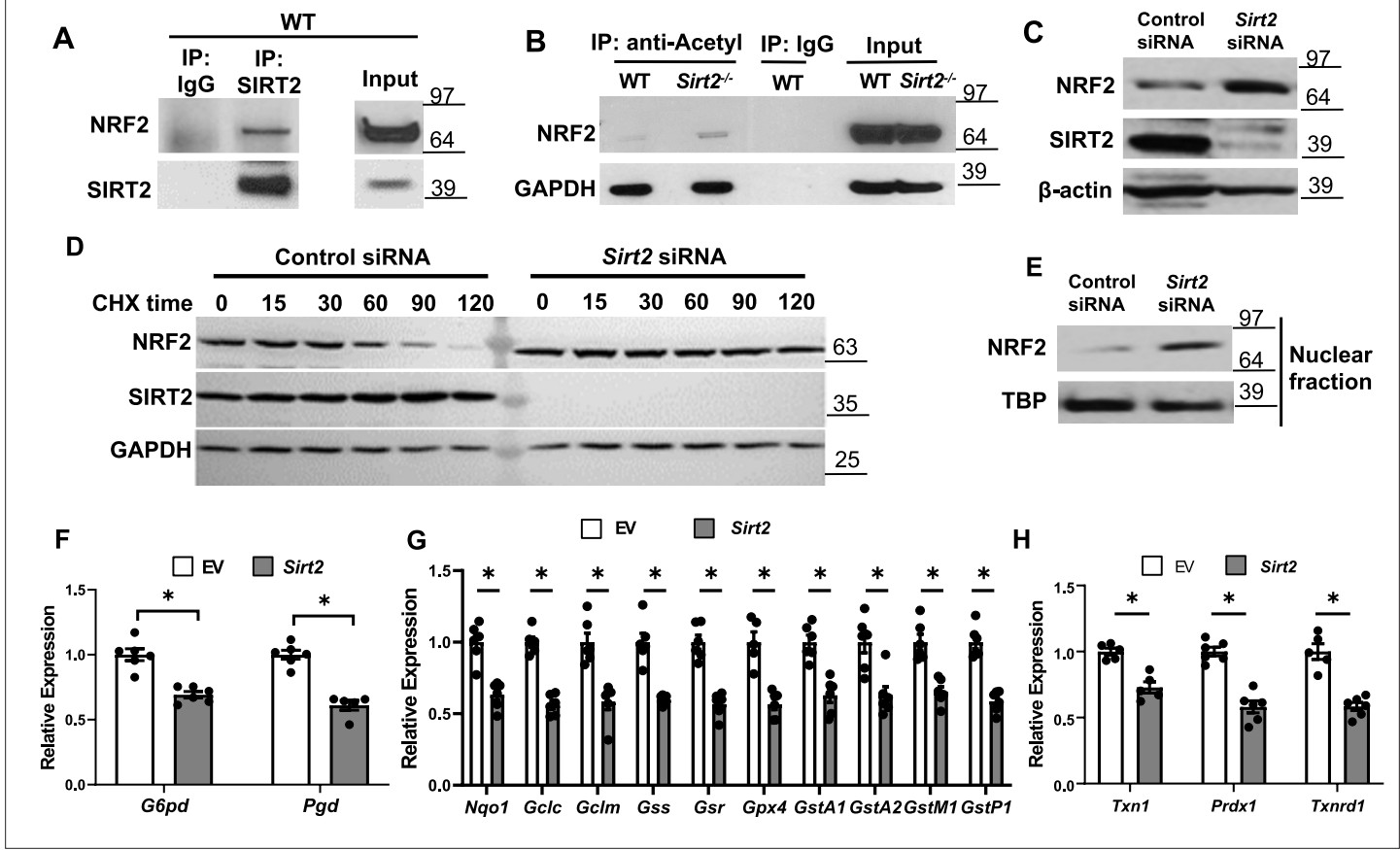

**Figure 5.** SIRT2 interacts with nuclear factor (erythroid-derived 2)-like 2 (NRF2) and regulates its activity in the heart. (**A**) Co-immunoprecipitation (IP) of SIRT2 and NRF2 in extracts of hearts from wild-type (WT) mice. (**B**) Endogenous NRF2 acetylation levels in the hearts of WT and *Sirt2*⁻/⁻ mice at the baseline. Acetylated proteins were IPed by anti-acetyl antibody followed by immunoblotting with anti-NRF2 antibody. (**C**) NRF2 protein levels in neonatal rat cardiomyocytes (NRCMs) treated with *Sirt2* siRNA. (**D**) NRF2 protein levels in H9c2 cells treated with control or *Sirt2* siRNA and harvested at different time points after treatment with 100 µg/ml of CHX. (**E**) NRF2 protein levels in the nucleus in NRCMs treated with control or *Sirt2* siRNA. (**F–H**) mRNA levels of NRF2 target genes in pentose phosphate pathway (**F**), quinone and glutathione-based detoxification (**G**), thioredoxin production (**H**) in H9c2 cells overexpressing empty vector (white bars) or SIRT2 (gray bars). *p<0.05 by Student's t-test.

The online version of this article includes the following source data and figure supplement(s) for figure 5:

**Source data 1.** mRNA with overexpression of EV or SIRT2 as shown in *Figure 5F*.

**Source data 2.** mRNA with overexpression of EV or SIRT2 as shown in *Figure 5G*.

**Source data 3.** mRNA with overexpression of EV or SIRT2 as shown in *Figure 5H*.

**Source data 4.** Uncropped gels for *Figure 5*.

**Source data 5.** Uncropped gels for *Figure 5* unedited.

**Source data 6.** Uncropped gels for *Figure 5* unedited.

**Figure supplement 1.** Effects of SIRT2 overexpression on mRNA levels of non-nuclear factor (erythroid-derived 2)-like 2 (NRF2) targeted antioxidant genes.

**Figure supplement 1—source data 1.** mRNA with overexpression of EV or SIRT2 as shown in *Figure 5—figure supplement 1*.

**Figure supplement 2.** SIRT2 regulates nuclear factor (erythroid-derived 2)-like 2 (NRF2) and its target proteins.

**Figure supplement 2—source data 1.** Uncropped gels for *Figure 5—figure supplement 2A*.

**Figure supplement 2—source data 2.** Uncropped gels for *Figure 5—figure supplement 2A* unedited.

**Figure supplement 2—source data 3.** mRNA with overexpression of EV or SIRT2 as shown in *Figure 5—figure supplement 2B–D*.

**Figure supplement 3.** Reactive oxygen species (ROS) levels as assessed by dihydroethidium (DHE) staining in neonatal rat cardiomyocytes (NRCMs) treated with control or *Sirt2* siRNA after treatment with 500 µM H₂O₂.

**Figure supplement 3—source data 1.** Fluorescence of cell death for *Figure 5—figure supplement 3*.

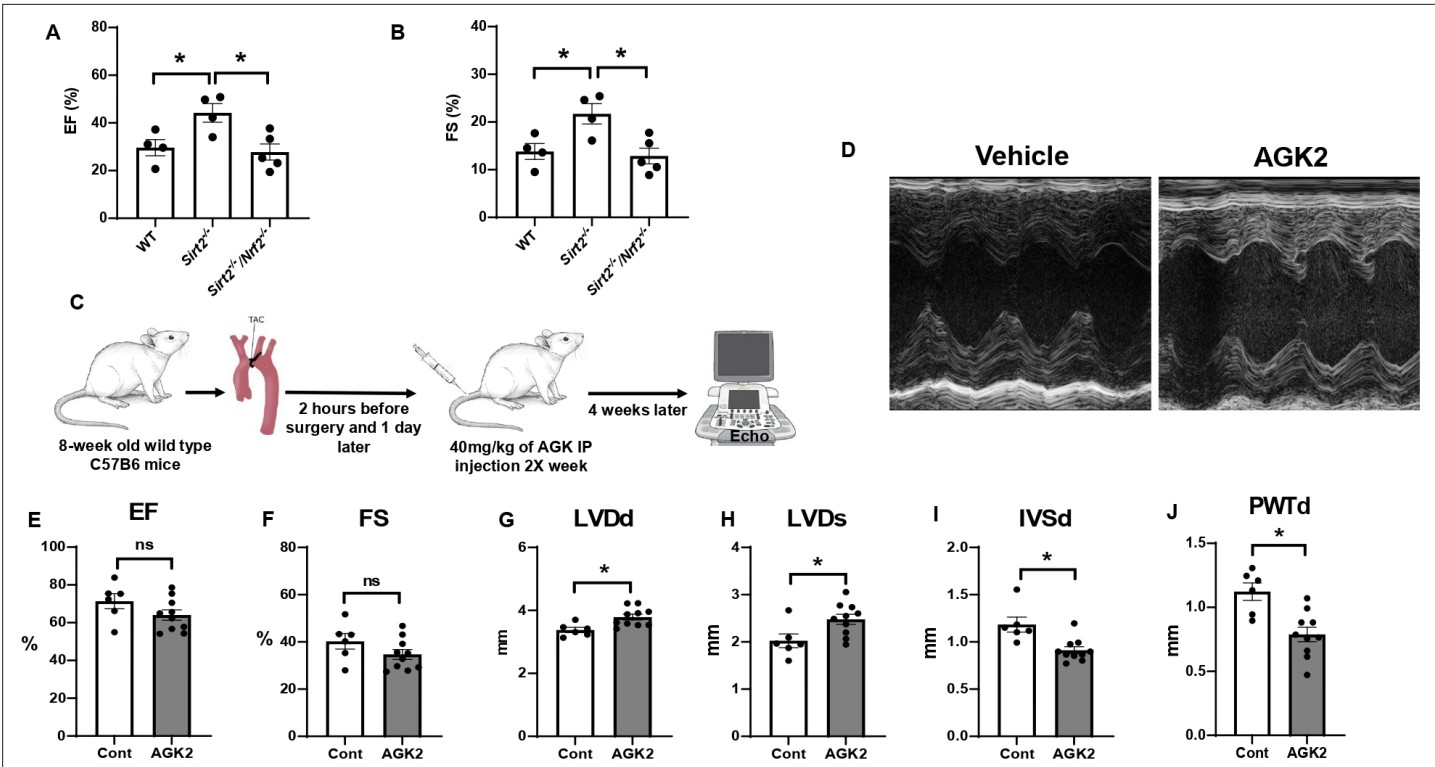

**Figure 6.** *Nrf2* deletion and SIRT2 inhibitors protected against cardiac damage and cardiac hypertrophy. Ejection fraction (EF) (**A**) and fractional shortening (FS) (**B**) in wild-type (WT), *Sirt2⁻/⁻*, and *Sirt2⁻/⁻/Nrf2⁻/⁻* double knockout (KO) mice 28 days after ischemia-reperfusion (I/R) (N=4–5). (**C**) Protocol for treatment of mice with SIRT2 inhibitor, AGK2. (**D**) Echo images of hearts from WT mice treated with either vehicle or AGK2. (**E–J**) EF (**E**), FS (**F**), left ventricular diameter during diastole (LVDd) (**G**), left ventricular diameter during systole (LVDs) (**H**), IVSd (**I**), and posterior wall thickness during diastole (PWTd) (**J**) in WT mice treated with AGK after trans-aortic constriction (TAC) according to the protocol in panel C (N=6–10). *p<0.05 by ANOVA for panels A–B or Student's t-test for panels E–J.

The online version of this article includes the following source data for figure 6:

**Source data 1.** Ejection fraction (EF) in wild-type (WT), *Sirt2⁻/⁻*, and *Sirt2⁻/⁻/Nrf2⁻/⁻* mice after ischemia-reperfusion (I/R) as shown in *Figure 6A*.

**Source data 2.** Fractional shortening (FS) in wild-type (WT), *Sirt2⁻/⁻*, and *Sirt2⁻/⁻/Nrf2⁻/⁻* mice after ischemia-reperfusion (I/R) as shown in *Figure 6B*.

**Source data 3.** Ejection fraction (EF) with AGK2 as shown in *Figure 6E*.

**Source data 4.** Fractional shortening (FS) with AGK2 as shown in *Figure 6F*.

**Source data 5.** Left ventricular diameter during diastole (LVDd) with AGK2 as shown in *Figure 6G*.

**Source data 6.** LVDs with AGK2 as shown in *Figure 6H*.

**Source data 7.** IVSd with AGK2 as shown in *Figure 6I*.

**Source data 8.** Posterior wall thickness during diastole (PWTd) with AGK2 as shown in *Figure 6J*.

## Pharmacological inhibition of SIRT2 protects the heart against ischemic damage

Since a reduction in SIRT2 levels led to protection against the development of HF and cardiac hypertrophy, we next studied whether pharmacological inhibition of SIRT2 also exerts protective effects in the heart in response to PO. For these studies, we used AGK2, a selective SIRT2 inhibitor (*He et al., 2012*; *Outeiro et al., 2007*; *Petrilli et al., 2013*). Eight-week-old C57B6 mice were underwent TAC and 1 day later, they were randomized to treatment with 40 mg/kg of AGK2 or vehicle intraperitoneally twice a week for 4 weeks. At the conclusion of the study, their cardiac function and heart chamber size were assessed using echocardiography (*Figure 6C*). Treatment with AGK2 did not change the systolic function of the heart, as assessed by EF, FS (*Figure 6D–F*). However, measures of cardiac size, as assessed by left ventricular (LV) diameter during diastole and systole (LVDd and LVDs, respectively), were increased, while measures of LV wall diameter, as assessed by IVSd and posterior wall thickness during diastole (PWTd), were reduced (*Figure 6G–J*). These results indicate that pharmacological

inhibition of SIRT2 can protect the heart against cardiac hypertrophy and improve cardiac remodeling in response to PO.

## Discussion

Sirtuins play a major role in post-translational modification of proteins, and their deletion have been shown to lead to a number of physiological changes and pathological conditions (*Baur et al., 2012*; *Watroba and Szukiewicz, 2021*; *Zhao et al., 2020*). Although multiple sirtuins have been investigated in the context of cardiovascular diseases (*Tang et al., 2017*; *Yuan et al., 2015*; *Sarikhani et al., 2018*), it is not known whether SIRT2 has a role in protection against HF and cardiac hypertrophy. In this paper, we used genetic models to show that SIRT2 has detrimental effects in the heart in the setting of cardiac insults and demonstrate that the deleterious effects of SIRT2 is through increased NRF2 deacetylation and its degradation and eventual reduction in the levels of antioxidant genes. We also show that deletion of *Nrf2* reverses the protective effects of *Sirt2* deletion. Finally, we provide a clinical significance for our findings and show that treatment of mice with AGK2, a selective SIRT2 inhibitor, results in protection against cardiac hypertrophy in response to PO.

NRF2 plays a major role in the regulation of genes involved in oxidative stress, metabolic processes, drug metabolism, and stress response, among others. Thus, its activation has been studied extensively in a number of diseases, however, these studies have not led to an effective therapy. It is possible that direct activation of NRF2 may have unwanted side effects. Our studies provide a proof of concept that targeting SIRT2 and indirect activation of NRF2 by altering its post-translational protein modification may prove to be a more effective therapeutic strategy for a number of diseases.

Cardiac hypertrophy is a major complication of hypertrophy and metabolic disorders in this country (*Husser et al., 2018*; *Moon et al., 2020*; *Semsarian et al., 2015*), however, our treatment options are limited. We generally treat the underlying cause without directly targeting cardiac function and remodeling. Our data indicate that targeting SIRT2 with AGK2 would improve cardiac remodeling and cardiac hypertrophy in response to PO, potentially providing a novel therapy for these disorders. However, systemic inhibition of SIRT2 may have unwanted side effects, which would need to be investigated further.

A few limitations should be taken into consideration regarding our findings. First, as global *Nrf2* KO mice were used in this study, we cannot rule out possible role of *Nrf2* deletion in non-cardiac tissues in reversing the effects of *Sirt2* deletion. Additionally, the systemic and cardiac-independent effects of AGK drug cannot be ruled out in our studies. Finally, our data are not consistent with two previous studies that have shown protective effects of SIRT2 in the heart, with one study showing that deletion of *Sirt2* increases age-related and angiotensin II-mediated cardiac hypertrophy (*Tang et al., 2017*), while another study showing that *Sirt2* deficiency leads to cardiac dysfunction and cardiac hypertrophy (*Sarikhani et al., 2018*). The reason for this discrepancy in our data is not clear, however, we used both global and cardiac specific KO of *Sirt2*, while these studies have used mice with global deletion of the gene. The genetic background of the mice and the different gene targeting strategy might have also contributed to the difference. Additionally, these studies used either angiotensin- or isoproterenol-induced models to cause cardiac hypertrophy and other potential effects of these drugs may explain the differences in our results. However, we used two different genetic models (global and cardiac specific KO of *Sirt2*) and a pharmacological approach to test our hypothesis, all of which produced similar results. Finally, we provide a mechanism for the deleterious effects of SIRT2 in the heart through its regulation of antioxidant proteins by NRF2 protein.

In summary, our data demonstrate that SIRT2 has deleterious effects in the heart through its post-translational modification of NRF2. We show that SIRT2 binds to NRF2 and that *Sirt2* deletion leads to an increase in NRF2 stability and nuclear levels, resulting in higher production of antioxidant genes. Additionally, deletion of *Nrf2* reverses the protective effects of *Sirt2* deletion. Finally, our results provide a potential therapy for cardiac hypertrophy by using AGK2, a specific inhibitor of SIRT2.

# Materials and methods

## Animal models

All animals were maintained and handled in accordance with the Northwestern Animal Care and Use Committee. $Sirt2^{-/-}$ and $Sirt2$ floxed mice were obtained from Dr. David Gius. $Nrf2^{-/-}$ mice were purchased from Jackson labs. $Sirt2^{-/-}/Nrf2^{-/-}$ mice were generated by crossing the $Sirt2^{-/-}$ with $Nrf2^{-/-}$ mice. All animals were kept in accordance with standard animal care requirements and maintained in a 22°C room with a 12 hr light/dark cycle, and received food and drinking water ad libitum.

## Study approval

All animal studies were approved by the Institutional Animal Care and Use Committee at Northwestern University (Chicago, IL, USA) and were performed in accordance with the guidelines from the National Institutes of Health. The approval number of the animal protocol currently associated with this activity is IS00006808.

## Human heart samples

Non-failing and cardiomyopathy cardiac tissue samples were obtained from the Human Heart Tissue Collection at the Cleveland Clinic. Informed consent was obtained from all the transplant patients and from the families of the organ donors before tissue collection. Protocols for tissue procurement were approved by the Institutional Review Board of the Cleveland Clinic (Cleveland, OH, USA), which is AAHRPP accredited. The experiments conformed to the principles set out in the WMA Declaration of Helsinki and the Department of Health and Human Services Belmont Report.

## Animal surgeries

Cardiac surgeries were performed as previously described (*Wu et al., 2011*; *Wu et al., 2012*). Briefly, mice of 10–12 weeks of age were anesthetized with 2% isoflurane. The animals were placed in a supine position and ECG leads were attached. The body temperature was monitored using a rectal probe and was maintained at 37°C with heating pads throughout the experiment. A catheter was inserted into the trachea and was then attached to the mouse ventilator via a Y-shaped connector. The mice were ventilated at a tidal volume of 200 µl and a rate of 105 breaths/min using a rodent ventilator. Chronic PO was induced by TAC as described. A 7-0 silk suture was placed around the transverse aorta between the origin of the right innominate and left common carotid arteries against an externally positioned 27-gauge needle to yield a narrowing 0.4 mm in diameter when the needle was removed after ligation. The sham procedure was identical except that the aorta was not ligated. For cardiac I/R injury, a 1 mm section of PE-10 tubing was placed on top of left anterior descending artery, and a knot was tied on the top of the tubing to occlude the coronary artery with an 8-0 silk suture. Ischemia was verified by pallor of the anterior wall of the left ventricle and by ST-segment elevation and QRS widening on the ECG. After occlusion for 45 min, reperfusion occurred by cutting the knot on top of the PE-10 tubing. Animals were given buprenorphine for post-operative pain.

## Echocardiography

Parasternal short- and long-axis views of the heart were obtained using a Vevo 770 high-resolution imaging system with a 30 MHz scan head. 2D and M-mode images were obtained and analyzed. EF was calculated from M-mode image using Teichholtz equation, and FS was directly calculated from end-systolic and end-diastolic chamber size from M-mode images.

## Histological analysis

Hearts were fixed in 10% formalin (PBS buffered), dehydrated, and embedded in paraffin. Heart architecture was determined from transverse 5 µm deparaffinized sections stained with H&E. Fibrosis was detected with Masson's trichrome staining.

## Cell culture and reagents

NRCMs were isolated from 1- to 2-day-old Sprague-Dawley rats as previously described (*Ichikawa et al., 2012*). Cardiomyocytes were cultured in DMEM, supplemented with 5% FBS, 1.5 mM vitamin B12, and 1 mM penicillin-streptomycin (Gibco). To prevent proliferation of non-myocytes, 100 µM

bromodeoxyuridine was added to the culture media. To induce oxidative stress, cells were exposed to hydrogen peroxide ($H_2O_2$, VWR) for 4 hr. CHX (Sigma) was used to check protein stability. H9c2 line were grown on DMEM, supplemented with 10% FBS. All cells were maintained in a 37°C incubator with 5% $CO_2$ and 6% oxygen and were 70–90% confluent when collected for various analyses unless otherwise noted. HL1 cells were cultured in Claycomb media (Sigma-Aldrich) containing 10% serum, 0.1 mM norepinephrine, 2 mM L-glutamine, and penicillin/streptomycin, as previously described (*Rines et al., 2017*). H9c2 cells were obtained from ATCC and were grown in Dulbecco's Modified Eagle's Medium (Corning) supplemented with 10% FBS (Atlanta Biologicals), as described previously (*Sato et al., 2018*). We routinely check for mycoplasma in our lab.

## Plasmids and transfections

pcDNA vector containing SIRT2 and corresponding empty vector were gifts from Dr. David Guis. *Sirt2* siRNA were purchased from Dharmacon. For plasmid and siRNA transient transfection, cells were transfected using Lipofectamine 2000 Transfection Reagent (Invitrogen, Thermo Fisher Scientific) and Dharmafect transfection reagent (Horizon) according to the manufacturer's instructions.

## Protein stability assay

For protein stability studies, 100 µg/ml of CHX (Sigma) was added to H9c2 cells, and samples were isolated at 0, 15, 30, 60, 90, and 120 min after the addition of CHX. Samples were then run on a gel for western blot analysis.

## RNA isolation and qRT-PCR

RNA was isolated using RNA-STAT60 (Tel-Test) according to the manufacturer's instructions and subjected to DNAse I (Ambion) digestion to remove residual DNA. Purified RNA was then reverse transcribed with random hexamer and oligo-dT (*Yuan et al., 2015*) (Applied Biosystems) and amplified on a 7500 Fast Real-Time PCR system using Fast SYBR Green PCR Master Mix (Applied Biosystems). The sequences for primers are included in *Appendix 1—table 1*. mRNA levels were calculated based on the difference of threshold Ct values in the target gene and average Ct values of *18s, Actb, B2m,* and *Hprt* in the same sample.

## Cell death studies

Permeability to PI (Sigma-Aldrich) was used as a fluorescent signal for cell death (*Ardehali et al., 2005*). NRCMs were treated with $H_2O_2$ for 4 hr, washed one time with HBSS and co-stained with PI and Hoescht. After several steps of wash, images were taken using Zeiss AxioObserver.Z1 fluorescent microscope. Data were analyzed with ImageJ (NIH).

## Measurement of ROS

Intracellular ROS levels were determined using dihydroethidium (DHE) assay. Briefly, NRCMs were treated with $H_2O_2$. After 4 hr of incubation, cells were washed and loaded with 10 µM DHE and Hoechst for 30 min. After two washing steps, fluorescence images were acquired with the Zeiss Axio-Observer.Z1.

## Isolation of nuclei

Nuclei were isolated using NE-PER Nuclear and Cytoplasmic Extraction Reagents (Pierce).

## Western blot and IP

20–40 µg of protein were resolved on SDS-PAGE gels and transferred to nitrocellulose membranes (Invitrogen, CA, USA). The membranes were probed with antibodies against SIRT1, SIRT2 (Sigma-Aldrich), SIRT3, SIRT6 (Cell Signaling Technology), NRF2 (Abcam, cell signaling), HPRT, GAPDH (Proteintech), TBP, and β-actin (Abcam). HRP-conjugated donkey anti-rabbit and donkey anti-mouse were used as secondary antibodies (Jackson ImmunoResearch) and visualized by Pierce Super Signal Chemiluminescent Substrates.

For IP, cells or tissue were lysed using IP buffer (25 mM Tris-HCl pH 7.5, 150 mM NaCl, 1 mM EDTA, 0.1% NP-40, and 5% glycerol), and cell extracts were incubated overnight with appropriate antibodies

followed by incubation with protein A or G agarose beads for 4 hr at 4°C. After washing five times with IP buffer, immunocomplexes were resolved using SDS-PAGE and analyzed by western blot.

## Treatment of WT mice with AGK2

AGK2 administration to C57BL6J mice which underwent TAC surgery by intraperitoneal injection was started 1 day after the surgery with the dose of 40 mg/kg. The injection was performed twice a week for 4 weeks, and then their cardiac function was assessed by echocardiography.

AGK2 (Selleckchem) was dissolved in DMSO, and administered to C57BL6-J mice 2 hr before TAC surgery and then twice a week at a dose of 40 mg/kg started on day 1 after the surgery and continued for 2 weeks. At the end of studies, cardiac function was assessed by echocardiography.

## Statistical analysis

For sample size, based on our previous experience with similar studies, we estimated at least four to six animals per group are needed to detect significant functional difference. However, the sample size was not pre-determined. Animals were randomized into sham versus control group. Surgical operator was blinded regarding animal's genotype. For AGK2 treatment study, mice were randomized into control versus AGK2 group. Data analysis was not masked. The replicate number for in vitro experiment and animal numbers for in vivo experiments were based on our prior experience of studying gene expression and cardiac function after insult. All reported replicates for in vitro experiments are technical replicates (unless otherwise specifically noted).

All data are expressed as mean ± SEM. Exclusion criteria were not pre-established. No sample or data points were omitted from analysis. Statistical significance was assessed with two-tailed unpaired t-test for two group comparison or with ANOVA for data with more than two groups. Post hoc Tukey's test was performed for multiple-group comparison if ANOVA reached statistical significance. Kolmogorov-Smirnov test was used to test for normal distribution. Levene's test was used to evaluate equal variance among groups. A p-value of $<0.05$ was considered statistically significant.

## Acknowledgements

The authors like to thank David Gius and Athanassios Vassilopoulos for helpful discussion and providing mice and reagents. We would also like to thank Mingyang Liu for managing our mouse colony and genotyping the mice used in this study. We thank Chunlei Chen for conducting the animal surgery experiments. The authors also thank Dr. Sathyamangla Prasad from Cleveland Clinic Foundation for providing the human samples to us. XY is supported by the American Heart Association grant 14POST20490097. YT is supported by the American Heart Association grant POST000204352. HA is supported by NIH R01 HL140973, R01 HL138982, R01 HL140927, R01 HL155953, and a grant from Leducq foundation.

## Additional information

### Funding

| Funder | Grant reference number | Author |
| --- | --- | --- |
| National Institutes of Health | R01 HL140973 | Hossein Ardehali |
| National Institutes of Health | R01 HL138982 | Hossein Ardehali |
| National Institutes of Health | R01 HL140927 | Hossein Ardehali |
| National Institutes of Health | R01 HL155953 | Hossein Ardehali |
| Leducq | Cardiooncology Network | Hossein Ardehali |
| American Heart Association | 14POST20490097 | Xiaoyan Yang |

| Funder | Grant reference number | Author |
|---|---|---|
| American Heart Association | POST000204352 | Yuki Tatekoshi |

The funders had no role in study design, data collection and interpretation, or the decision to submit the work for publication.

## Author contributions

Xiaoyan Yang, Hsiang-Chun Chang, Data curation, Software, Formal analysis, Validation, Visualization, Methodology, Writing – review and editing; Yuki Tatekoshi, Data curation, Formal analysis, Validation, Investigation, Visualization, Methodology, Writing – review and editing; Amir Mahmoodzadeh, Data curation, Formal analysis, Project administration; Maryam Balibegloo, Rongxue Wu, Chunlei Chen, Tatsuya Sato, Investigation, Methodology; Zeinab Najafi, Data curation, Formal analysis; Jason Shapiro, Data curation, Formal analysis, Investigation, Methodology; Hossein Ardehali, Conceptualization, Resources, Data curation, Formal analysis, Supervision, Funding acquisition, Investigation, Writing – original draft, Project administration, Writing – review and editing

## Author ORCIDs

Xiaoyan Yang (ID) https://orcid.org/0000-0002-4450-7554
Amir Mahmoodzadeh (ID) http://orcid.org/0000-0002-0523-8152
Tatsuya Sato (ID) http://orcid.org/0000-0001-7876-1772
Jason Shapiro (ID) https://orcid.org/0000-0003-0880-3142
Hossein Ardehali (ID) https://orcid.org/0000-0002-7662-0551

## Ethics

All animals were maintained and handled in accordance with the Northwestern Animal Care and Use Committee. All animal studies were approved by the Institutional Animal Care and Use Committee at Northwestern University (Chicago, Illinois) and were performed in accordance with guidelines from the National Institutes of Health. The approval number of the animal protocol currently associated with this activity is IS00006808.

## Decision letter and Author response

Decision letter https://doi.org/10.7554/eLife.85571.sa1
Author response https://doi.org/10.7554/eLife.85571.sa2

# Additional files

## Supplementary files
• MDAR checklist

## Data availability
Source data files are provided.

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

# Appendix 1

**Appendix 1—table 1.** Primer sequences.

| Genes | Forward primer | Reverse primer | Species |
|---|---|---|---|
| *Nppa* | GGGTAGGATTGACAGGATTGG | CCTCCTTGGCTGTTATCTTC | Mouse |
| *18 s* | AGTCCCTGCCCTTTGTACACA | CGATCCGAGGGCCTCACTA | |
| *Actb* | CTAAGGCCAACCGTGAAAAG | ACCAGAGGCATACAGGGACA | Mouse |
| *Nppb* | ATCCGTCAGTCGTTTGGG | CAGAGTCAGAAACTGGAGTC | Mouse |
| *G6pd* | GGCCAACCGTCTGTTCTACCT | CCACTATGATGCGGTTCCAGC | Rat |
| *Pgd* | CGGGTCATACTGCTCGTGAA | AGGTCCTGGCATCTTCTTGTCG | Rat |
| *Nqo1* | CACTACGATCCGCCCCCAAC | GCGTGGGCCAATACAATCAGG | Rat |
| *Gclc* | GTCAAGGACCGGCACAAGGA | GAACATCGCCGCCATTCAGT | Rat |
| *Gclm* | TGCCACCAGATTTGACTGCATTT | TCCTGGAAACTTGCCTCAGAGAG | Rat |
| *Gss* | GAGGTCCGCAAAGAACCCCA | GAGCGTGAATGGGGCATACG | Rat |
| *Gsr* | TCACCCCGATGTATCACGCT | GCCCTGAAGCATCTCATCGC | Rat |
| *Gpx4* | AGCAACAGCCACGAGTTCCT | ATCGATGTCCTTGGCTGCGA | Rat |
| *Gsta1* | ACTTCGATGGCAGGGGGAGAA | TGGAACATCAAACTCCCATCATTCC | Rat |
| *Gsta2* | TTGACGGGATGAAGCTGGCA | GTCAGATCTAAAATGCCTTCGGTGT | Rat |
| *Gstm1* | CCAAGTGCCTGGACGCCTTC | ATAGGTGTTGAGAGGTAGCGGC | Rat |
| *Gstp1* | CGTCCACGCAGCTTTGAGT | GTAACCACCTCCTCCTTCCAGC | Rat |
| *Txn1* | AGTAGACGTGGATGACTGCCA | AGCACCAGAGAACTCCCCAAC | Rat |
| *Prdx1* | TCAGATCCCAAGCGCACCAT | AGCGGCCAACAGGAAGATCA | Rat |
| *Txnrd1* | AATGCTGGAGAGGTGACGCA | GATGTCTCCCCCAGAACGCT | Rat |
| *Sirt1* | CAGTGTCATGGTTCCTTTGC | CACCGAGGAACTACCTGAT | Mouse |
| *Sirt3* | GCTGCTTCTGCGGCTCTATAC | GAAGGACCTTCGACAGACCGT | Mouse |
| *Sirt4* | GTGGAAGAATAAGAATGAGCGGA | GGCACAAATAACCCCGAGG | Mouse |
| *Sirt5* | CCACCGACAGATTCAGGTTT | TTCCCGTTAGTGCCCTGCTTTA | Mouse |
| *Sirt6* | ATGTCGGTGAATTATGCAGCA | GCTGGAGGACTGCCACATTA | Mouse |
| *Sirt7* | CAGGTGTCACGCATCCTGAG | GCCCGTGTAGACAACCAAGT | Mouse |
| *Cat* | CCAGCCAGCGACCAGATGAA | CCTATTGGGTTCCCGCCTCC | Rat |
| *Sod1* | AACTGAAGGCGAGCATGGGTT | ATGCCTCTCTTCATCCGCTGG | Rat |
| *Sod2* | GGGGCCATATCAATCACAGCA | GAACCTTGGACTCCCACAGAC | Rat |
| *Sod3* | ACGTTCTTGGGAGAGCTTGT | CTGCTAAGTCGACACCGGAC | Rat |
| *Actb* | GGCTCCTAGCACCATGAAGA | CAGTGAGGCCAGGATAGAGC | Rat |
| *Hprt 1* | CCCTCAGTCCCAGCGTCGTG | CGAGCAAGTCTTTCAGTCCTGTCC | Rat |
| *B2m* | CCGTGATCTTTCTGGTGCTTG | GAGACACGTAGCAGTTGAGGA | Rat |
| *G6pd* | GTCTTTGCTCGGTGCTTGTC | AGCATAGAGGGCCTTACGGA | Mouse |
| *Nqo1* | TCTCTGGCCGATTCAGAGTG | CCAGACGGTTTCCAGACGTT | Mouse |
| *Gclm* | ATGACCCGAAAGAACTGCTC | TGGGTGTGAGCTGGAGTTAAG | Mouse |
| *Gclc* | ACTGAATGGAGGCGATGTTCTT | CAGAGGGTCGGATGGTTGG | Mouse |
| *Gss* | GCACCGACACGTTCTCAATG | TAGCACCACCGCATTAGCTG | Mouse |

*Appendix 1—table 1 Continued on next page*

*Appendix 1—table 1 Continued*

| Genes | Forward primer | Reverse primer | Species |
|-------|----------------|----------------|---------|
| Gsr | ATGTTGACTGCCTGCTCTGG | ATCCGTCTGAATGCCCACTT | Mouse |
| Gpx4 | GTACTGCAACAGCTCCGAGT | ATGCACACGAAACCCCTGTA | Mouse |
| Gsta2 | CCAGGACTCTCACTAGACCGT | CCCGGGCATTGAAGTAGTGA | Mouse |
| gstm1 | ATACACCATGGGTGACGCTC | TCCATCCAGGTGGTGCTTTC | Mouse |
| Gstp1 | GTCTACGCAGCACTGAATCC | GGGAGCTGCCCATACAGACA | Mouse |
| Txn1 | GCGCTCCGCCCTATTTCTAT | CCTCCTGAAAAGCTTCCTTGC | Mouse |
| Prdx1 | ACTGACAAACATGGTGAAGTGTG | TACAAGAGTTTCTTCTGGCTGC | Mouse |
| Txnrd1 | GAATGGACAGTCCCATCCCG | AAGCCCACGACACGTTCATC | Mouse |
| Actb | TAAAACCCGGCGGCGCA | GTCATCCATGGCGAACTGGT | Mouse |
| Hprt1 | AGAGCGTTGGGCTTACCTCA | TGGTTCATCATCGCTAATCACG | Mouse |
| B2m | ACGCCTGCAGAGTTAAGCAT | TGATCACATGTCTCGATCCCAG | Mouse |

## Appendix 1—key resources table

| Reagent type (species) or resource | Designation | Source or reference | Identifiers | Additional information |
|---|---|---|---|---|
| Strain, strain background (*Mus musculus*, male, C57BL/6) | Sirt2 knockout mice | Dr. Gius Lab | | refer to: SIRT2 Maintains Genome Integrity and Suppresses Tumorigenesis through Regulating APC/C Activity. |
| Strain, strain background (*Mus musculus*, male, C57BL/6) | Sirt2 flox/flox mince | Dr. Gius Lab | | refer to: SIRT2 deletion enhances KRAS-induced tumorigenesis in vivo by regulating K147 acetylation status |
| Strain, strain background (*Mus musculus*, male, C57BL/6) | Nrf2 knockout mice | the Jackson Laboratory | RRID:IMSR_JAX:017009 | |
| Strain, strain background (*Rattus norvegicus domestica*, female) | Sprague–Dawley rat | Charles River | | |
| Antibody | Rabbit polyclonal SIRT1 antibody | Sigma | 07–131 | WB (1:1000) |
| Antibody | Rabbit polyclonal SIRT2 antibody | Sigma | S8447 | WB (1:1000) |
| Antibody | Rabbit mAb SIRT3 antibody | Cell Signaling Technology | Rabbit mAb #5490 | WB (1:1000) |
| Antibody | Rabbit polyclonal anti-HPRT antibody | Proteintech | 150-59-1-AP | WB (1:5000) |
| Antibody | Mouse monoclonal anti-GAPDH antibody | Proteintech | 60004–1-Ig | WB (1:10000) |
| Antibody | Rabbit mAb NRF2 antibody | Cell Signaling Technology | Rabbit mAb #20733 | WB (1:1000) |
| Antibody | Rabbit polycloncal NRF2 antibody | Abcam | ab31163 | WB (1:1000) |
| Antibody | HRP-conjugated donkey polyclonal anti-mouse IgG antibody | Jackson ImmunoResearch | 715-035-150 | WB (1:5000) |
| Antibody | HRP-conjugated donkey polyclonal anti-rabbit IgG antibody | Jackson ImmunoResearch | 711-035-152 | WB (1:5000) |
| Antibody | Rabbit Anti beta Actin antibody | Abcam | ab8227 | WB (1:2000) |

*Appendix 1 Continued on next page*

*Appendix 1 Continued*

| Reagent type (species) or resource | Designation | Source or reference | Identifiers | Additional information |
|---|---|---|---|---|
| Antibody | RabbTBP antibody | Abcam | ab63766 | WB (1:2000) |
| Antibody | Rabbit mAb SIRT6 antibody | Cell Signaling Technology | Rabbit mAb #12486 | WB (1:1000) |
| Antibody | Mouse Flag-M2 monoclonal antibody | Sigma | F1804 | WB: (1:2000); IP(1:xxx) |
| Antibody | Acetyl Lysine Antibody, Agarose | immunechem | ICP0388-2MG | IP: 1:10 |
| Chemical compound, drug | Propodium Iodine | Sigma | P4170-10MG | |
| Chemical compound, drug | Hoechst 34432 | Life Technology | 62249 | |
| Chemical compound, drug | Cycloheximide | Sigma | 1810 | |
| Chemical compound, drug | Paraformaldehyde | Thermo Fisher Scientific | AC416780250 | |
| Chemical compound, drug | RIPA Buffer | Thermo Fisher Scientific | 89901 | |
| Chemical compound, drug | ProteaseArrest Protease Inhibitor | G-Biosciences | 786–437 | |
| Chemical compound, drug | 10% formalin | Fisher Scientific | FLSF1004 | |
| Chemical compound, drug | BrdU | Sigma | 19–160 | |
| Chemical compound, drug | vitamin B12 | Sigma | V2876 | |
| Chemical compound, drug | FBS | Bio-Techne | S11550 | |
| Chemical compound, drug | penicillin–streptomycin | Cytiva | SV30010 | |
| Chemical compound, drug | Lipofectamine 2000 Transfection Reagent | Invitrogen, Thermo Fisher Scientific | 11668027 | |
| Chemical compound, drug | Dharmafect transfection reagent | Horizon | 2001–03 | |
| Chemical compound, drug | AGK2 | Selleckchem | S7577 | |
| Chemical compound, drug | hydrogen peroxide | Fisher Scientific | H324-500 | |
| Chemical compound, drug | Normal Rabbit IgG | Sigma | 12–370 | |
| Chemical compound, drug | DMSO | Sigma | D4540 | |
| Cell line | H9c2 | ATCC | CRL-1446 | |
| Other | DMEM | Corning | 10-013CV | |
| Commercial assay or kit | RNA-STAT60 | Teltest | Cs-502 | |
| Commercial assay or kit | DNAse I | Ambion | AM2222 | |
| Commercial assay or kit | PerfeCTa SYBR Green FastMix | Quanta | 95074–05 K | |
| Commercial assay or kit | qScript cDNA Synthesis Kit | Quanta | 95047–500 | |
| Commercial assay or kit | SuperSignal West Pico PLUS Chemiluminescent Substrate | Pierce | 34579 | |
| Commercial assay or kit | BCA Protein Assay Kit | Pierce | 23225 | |
| Commercial assay or kit | NE-PER Nuclear and Cytoplasmic Extraction Reagents | Pierce | PI78835 | |
| Commercial assay or kit | Trichrome Stain (Masson) Kit | Sigma | HT15-1KT | |

*Appendix 1 Continued on next page*

*Appendix 1 Continued*

| Reagent type (species) or resource | Designation | Source or reference | Identifiers | Additional information |
|---|---|---|---|---|
| Commercial assay or kit | Protein A Agarose | Roche | 11719408001 | |
| Commercial assay or kit | dihydroethidium (DHE) assay | Thermo Fisher Scientific | D11347 | |
| Sequence-based reagent | Rat Sirt2 siRNA | Horizon Discovery | M-082072-01-0005 | siGENOME Rat Sirt2 (361532) siRNA |
| Recombinant DNA reagent | Wildtype SIRT2 plasmid | Dr. Gius Lab | | refer to: SIRT2 Maintains Genome Integrity and Suppresses Tumorigenesis through Regulating APC/C Activity |
| Software, algorithm | GraphPad Prism | GraphPad | Version 9 | |
| Software, algorithm | ImageJ | NIH | 1.53 c | |

