## [Editor Report]

In this manuscript, the authors examine the role of Sirt2 on cardiac hypertrophy by using 2 in-vivo models- systemic KO of Sirt2 and cardiac specific KO of Sirt 2. They have shown that Sirt2 is important for development of heart failure and cardiac hypertrophy. Mechanistically, the authors show that Sirt2 regulates NRF2 and that deletion of Sirt2 is protective through stabilization and increased nuclear translocation of NRF2.

---

## [Decision Letter]

**Decision letter after peer review:**

Thank you for submitting your article "SIRT2 inhibition protects against cardiac hypertrophy and heart failure" for consideration by *eLife*. Your article has been reviewed by 2 peer reviewers, and the evaluation has been overseen by a Reviewing Editor and Mone Zaidi as the Senior Editor. The reviewers have opted to remain anonymous.

Essential revisions:

1. The title and abstract should be modified to reflect the involvement of Sirt2 during I/R as well, as hypertrophy.

2. Figure 3 – did KO of Sirt2 in mice have any effect on infarct size?

3. Do any of the other Sirt proteins i.e. Sirt 1/3 levels change in the Sirt 2 cardiac-specific Sirt2 KO mouse?

4. Figure 4 shows that cKO Sirt2 mice are protected against TAC. What does the cardiac function of these mice look like following I/R?

5. Figure 5B – looks like the blot was generated from 2 separate blots. Please provide a better quality of western.

6. Figure supplemental 1A- please improve the quality of the western blot.

7. Gender of mice: The study was mainly done in male mice; however, it will be helpful to show similar effects in female mice.

8. The study was done in rat cell lines, it would be helpful to use mouse cell lines to draw the comparisons.

9. The authors should report some of the study limitations in greater detail, especially the contradictory results obtained by previous reports.

---

## [Author Response]

Essential revisions:1. The title and abstract should be modified to reflect the involvement of Sirt2 during I/R as well, as hypertrophy.

Thank you for this suggestion. The changes are made to the manuscript.

2. Figure 3 – did KO of Sirt2 in mice have any effect on infarct size?

We have analyzed the degree of damage to the heart in *Sirt2*^f/f^ and in cs-*Sirt2*^-/-^ mice and show in Figure 4F that the necrotic area (representing the degree of ischemic damage) is smaller in the cs-*Sirt2*^-/-^ mice.

3. Do any of the other Sirt proteins i.e. Sirt 1/3 levels change in the Sirt 2 cardiac-specific Sirt2 KO mouse?

We have assessed the levels of SIRT1/3 in cs-*Sirt2*^-/-^ hearts and show that the levels of these proteins do not change, as shown below and in Figure 4- figure supplement 1.

4. Figure 4 shows that cKO Sirt2 mice are protected against TAC. What does the cardiac function of these mice look like following I/R?

The hearts of these mice are also protected against I/R. The data is included in Figure 4E.

5. Figure 5B – looks like the blot was generated from 2 separate blots. Please provide a better quality of western.

This image is from one blot. The same membrane was first blotted with anti-NRF2, stripped and then blotted with anti-GAPDH. In the full gel file, we show that the blot is not from the same gel.

6. Figure supplemental 1A- please improve the quality of the western blot.

A new blot is now provided.

7. Gender of mice: The study was mainly done in male mice; however, it will be helpful to show similar effects in female mice.

Thank you. Our studies were not done only in male mice. However, as the Reviewer had indicated, we performed TAC studies in female mice with cardiac specific deletion of *Sirt2* and the results show similar effects. The results are now shown in Figure 4—figure supplement 2.

8. The study was done in rat cell lines, it would be helpful to use mouse cell lines to draw the comparisons.

We have performed a number of studies in mouse cell lines, including the data presented in Figures 5C, 5F, 5G and 5H. These new studies are now shown in Figure 5—figure supplement 1.

9. The authors should report some of the study limitations in greater detail, especially the contradictory results obtained by previous reports.

This is now added to the Discussion section.